# TRIM28 Regulates Dlk1 Expression in Adipogenesis

**DOI:** 10.3390/ijms21197245

**Published:** 2020-09-30

**Authors:** Hsin-Pin Lu, Chieh-Ju Lin, Wen-Ching Chen, Yao-Jen Chang, Sheng-Wei Lin, Hsin-Hui Wang, Ching-Jin Chang

**Affiliations:** 1Graduate Institute of Biochemical Sciences, College of Life Science, National Taiwan University, Taipei 10617, Taiwan; phoebe1620@gmail.com (H.-P.L.); ma8374@hotmail.com (C.-J.L.); a0972718683@gmail.com (W.-C.C.); 2Institute of Biological Chemistry, Academia Sinica, Taipei 11529, Taiwan; ntugeorge@gmail.com (Y.-J.C.); sanway@gate.sinica.edu.tw (S.-W.L.); 3Department of Pediatrics, Division of Pediatric Immunology and Nephrology, Taipei Veterans General Hospital, Taipei 11217, Taiwan; hhwang@vghtpe.gov.tw; 4Department of Pediatrics, Faculty of Medicine, School of Medicine, National Yang-Ming University, Taipei 11217, Taiwan; 5Institute of Emergency and Critical Care Medicine, School of Medicine, National Yang-Ming University, Taipei 11217, Taiwan

**Keywords:** TRIM28, Dlk1, adipogenesis, DNA methylation, histone modification

## Abstract

The tripartite motif-containing protein 28 (TRIM28) is a transcription corepressor, interacting with histone deacetylase and methyltransferase complexes. TRIM28 is a crucial regulator in development and differentiation. We would like to investigate its function and regulation in adipogenesis. Knockdown of Trim28 by transducing lentivirus-carrying shRNAs impairs the differentiation of 3T3-L1 preadipocytes, demonstrated by morphological observation and gene expression analysis. To understand the molecular mechanism of Trim28-mediated adipogenesis, the RNA-seq was performed to find out the possible Trim28-regulated genes. Dlk1 (delta-like homolog 1) was increased in Trim28 knockdown 3T3-L1 cells both untreated and induced to differentiation. *Dlk1* is an imprinted gene and known as an inhibitor of adipogenesis. Further knockdown of Dlk1 in Trim28 knockdown 3T3-L1 would rescue cell differentiation. The epigenetic analysis showed that DNA methylation of Dlk1 promoter and differentially methylated regions (DMRs) was not altered significantly in Trim28 knockdown cells. However, compared to control cells, the histone methylation on the *Dlk1* promoter was increased at H3K4 and decreased at H3K27 in Trim28 knockdown cells. Finally, we found Trim28 might be recruited by transcription factor E2f1 to regulate *Dlk1* expression. The results imply Trim28-Dlk1 axis is critical for adipogenesis.

## 1. Introduction

TRIM28 belongs to the tripartite motif family, containing N-terminal Ring finger, B boxes, and coiled-coil leucine zipper (RBCC) domain. RBCC is necessary for interaction with Krüppel-associated box (KRAB)-containing zinc-finger proteins (ZFPs) to silence genes and form oligomers [1,2,3,4]. The mechanism of TRIM28-mediated gene repression is through the recruitment of histone deacetylase (HDAC) complex NuRD (nucleosome remodeling deacetylase) and histone H3 lysine 9-specific methyltransferase SETDB1 by C-terminal PHD and bromodomain [5,6]. The central PxVxL pentapeptide associates with heterochromatin protein 1 (HP1) [7]. KRAB-ZFPs and TRIM28 complex–mediated chromatin DNA recognition was accompanied by DNA methylation [8]. TRIM28 is a crucial regulator in development and differentiation. The knockout of *Trim28* gene results in embryonic lethality indicating that it plays an essential role in embryonic development [9]. It is required for the maintenance and pluripotency of embryonic stem cells [10,11]. Recent reports showed that TRIM28 haploinsufficiency in both mouse and human leads to lean and obese phenotypes arising from the identical genotypes through dysregulation of an imprinted gene network [12,13]. 

Adipogenesis was extensively studied by using mouse preadipocyte cell line 3T3-L1 [14]. 3T3-L1 preadipocytes were growth-arrested followed by adding differentiation inducers to trigger adipogenesis. Most studies focused on transcriptional regulation and a series of transcription activators and repressors were found to play functions in this process [15]. Recently, epigenetic regulation in adipogenesis was examined. Dynamic and distinct histone modifications were analyzed to understand the expression of key adipogenesis regulatory genes, indicating that H3K27 methyltransferase Ezh2 facilitates adipogenesis by repressing adipogenic inhibitors Wnt and Pref-1(preadipocyte factor-1) [16,17,18]. In addition, H3K9 methyltransferase G9a and SETDB1 inhibits adipogenesis but H3K4 demethylase LSD1 promotes adipogenesis through modifying PPARγ, C/EBPα, and Wnts promoters [19,20,21,22,23]. Inhibition of DNA methylation can also activate Wnt10a to suppress adipogenesis in 3T3-L1 cells [24]. Constitutive Dlk1(also named Pref-1) expression or the addition of a soluble form of Dlk1 inhibits 3T3-L1 adipocyte differentiation, while downregulation of Dlk1 by antisense expression enhances adipogenesis [25,26,27,28,29,30]. Dlk1 has been reported to be expressed from the paternal allele, but not from the maternal allele, which is controlled by DNA methylation on differentially methylated regions (DMRs) [31,32,33]. 

To explore the functional mechanism of TRIM28-mediated cellular differentiation, we knocked down Trim28 in 3T3-L1 preadipocytes. After RNA-seq analysis, the possible Trim28 targets were selected for further investigation. We found Trim28 regulates Dlk1 expression through modulation of histone methylations. 

## 2. Results

### 2.1. Knockdown of Trim28 Impairs Adipogenesis

To investigate the functional effect of Trim28 in adipogenesis, we knocked down Trim28 in 3T3-L1 preadipocytes by using Letivirus-carrying shRNAs. As shown in Figure 1a, both shRNAs decreased Trim28 expression in mRNA and protein levels. The cells were induced to differentiation by FDMI (FBS, DEX, MIX and Insulin) treatment (see materials and methods). Oil red O staining showed knockdown (KD) of Trim28 decreased lipid formation significantly (Figure 1b). Consistent with the phenotype, the mRNA levels of differentiation-related transcriptional factors and the adipocyte markers also were downregulated in Trim28 KD cells (Figure 1c). In Figure 1d, cell numbers of Trim28 KD cells were decreased after induced to differentiation. The results indicate Trim28 plays a functional role during adipogenesis.

### 2.2. RNA-Seq Analysis Identifies the Possible Targets of Trim28 in Adipogenesis

To understand the molecular mechanism of Trim28-mediated adipogenesis, total RNAs were isolated from control and Trim28 KD cells without treatment or with inducing differentiation for two days for RNA-seq analysis. Downstream analyses include gene expression and deep analysis based on gene expression involving gene ontology (GO) enrichment analysis and Kyoto encyclopedia of genes and genomes (KEGG) pathway enrichment analysis. Pathway analysis of differentially expressed genes revealed that GO terms were associated with cell cycle, fibroblast proliferation, and chromatin silencing at day 0, and fat cell differentiation, fibroblast proliferation, and cell cycle arrest at day 2 (Figure 2a, Appendix A). These data indicated KD of Trim28 might reduce the proliferation of 3T3-L1, resulting in suppression of adipogenesis. In KEGG analysis, differentially expressed genes associated with cell cycle at day 0 were highly enriched in the upregulated gene set, and PPAR signaling pathway at day 2 were significantly downregulated in Trim28 KD cells (Figure 2b). The detailed genes in each pathway analysis were classified with upregulation and downregulation shown in Appendix A. In Figure 2c, the expression levels of TRIM28, C/EBPα, C/EBPβ, PPARγ, and Fabp4 (Ap2) in fragments per kilobase of transcript per million mapped reads (FPKM) were similar with qPCR analysis in Figure 1c. 

### 2.3. Dlk1 Is Upregulated in TRIM28 KD 3T3-L1 Preadipocyte

Because TRIM28 is a corepressor and Trim28 KD in 3T3-L1 results in suppression of adipogenesis, we considered TRIM28 might regulate the expression of repressors instead of activators in 3T3-L1. In FPKM data, we found the expression levels of Dlk1 were significantly increased in FPKM at day 0 and day 2, and the relative mRNA levels during adipogenesis were also increased in Trim28 KD cells (Figure 3a,b). We further performed chromatin immunoprecipitation assay by using anti-Trim28 to explore whether Trim28 targets to *Dlk1* in 3T3-L1. In Figure 3c, TRIM28 directly binds to the *Dlk1* promoter under non-induced condition (day 0). 

### 2.4. Knockdown of Dlk1 Rescues the Cell Differentiation in Trim28 KD Cells

To examine whether TRIM28 regulates adipogenesis through suppression of Dlk1 expression, we created double-knockdown cells. The knockdown efficiency of *Dlk1* and *Trim28* was confirmed by real-time PCR. The result indicated that the Dlk1 shRNA approximately decreased *Dlk1* expression by 99% and 61% in single and double knockdown cells relative to the shCtrl control (Figure 4a, left). And Trim28 was decreased by 90.8% and 66% in single and double knockdown cells (Figure 4a, right). The knockdown cells were triggered to differentiation, and the mRNA and protein expression levels of adipogenic transcription factor PPARγ were monitored by real-time PCR and western blot (Figure 4b,c). The expression of *Pparg* in preadipocytes is deficient, and induced in the second day of differentiation in shCtrl cells, whereas Trim28 KD suppressed its induction (Figure 4b). Knockdown of Dlk1 increased the mRNA levels of *Pparg* in day 0 and day 2 cells and it also restored the expression in Trim28 KD cells (Figure 4b). Consistent with the mRNA expression, PPARγ protein expression exhibited a similar regulation in knockdown of Trim28 or/and Dlk1 cells (Figure 4c). Moreover, we used Oil red O staining to measure adipogenesis. As shown in Figure 4d, the cells were induced to differentiation for eight days, and Dlk1 KD cells enhanced the adipogenesis with 2-fold of cells population differentiating into lipid-rounded adipocytes relative to control. In contrast, the rounded-lipid filled adipocytes significantly declined in knockdown of Trim28. Interestingly, Dlk1 and Trim28 double KD cells significantly restored adipogenesis (Figure 4d). Taken together, the effect of Dlk1 knockdown suggests that Dlk1 is a crucial downstream factor in Trim28-regulated adipogenesis.

### 2.5. Epigenetic Analysis of Dlk1 Regulatory Regions

Since TRIM28 is an epigenetic regulator, we examine the DNA methylation and histone modifications in the regulatory regions of *Dlk1* gene during adipogenesis. First, we performed bisulfite sequencing to analyze DNA methylation. There are 50, 24, 32, and 27 CpG dyads in *Dlk1* promoter, *Dlk1*-DMR, IG-DMR, and *Gtl2*-DMR, respectively (Figure 5a, Appendix A). In Figure 5b, we compared the day 0 (no induction) and day 2 (FDMI induction) cells. The DNA methylated status of *Dlk1* promoter was quite different from *Dlk1*-DMR, IG-DMR, and *Gtl2*-DMR, presenting in hypomethylated grade. Still, there are no significant differences in methylated quality between day 0 and day 2 cells at these regions after calculating methylated CpG numbers from 10–20 clones. However, the specific methylation sites at no.11 and no.20 CpG dyads in *Dlk1* promoter were observed in two-day differentiated cells. Moreover, we analyzed the DNA methylation in Trim28 KD 3T3-L1 cells compared to shCtrl cells in day 0 (Figure 5c) and day 2 sections (Figure 5d). In day 0 cells, although IG-DMR and *Dlk1* promoter presented slightly increased methylation in Trim28 KD cells compared to shCtrl cells, there are no significant differences after statistical analysis (Figure 5c, right panel). There are also no significant differences in four regulated regions among day 2 cells (Figure 5d, right panel). Taken together, the bisulfite sequencing analyses indicate that the DMRs of *Dlk1-Gtl2* gene cluster are hypermethylated and *Dlk1* promoter is hypomethylated. They were nearly not changed by knockdown of Trim28 and differentiation. 

Next, the histone modifications were examined. Because *Dlk1* expression in adipogenesis was reported to be repressed by Ezh2-mediated H3K27 methylation [18], we performed ChIP-qPCR to monitor the enrichment of H3K27me3 on *Dlk1* gene regulatory regions in shCtrl and Trim28 KD 3T3-L1 cells (Figure 6a). The active chromatin marker H3K4me3 was also observed (Figure 6b). Lower enriched levels of these two histone modifications on IG-DMR and *Gtl2*-DMR than on *Dlk1* promoter were detected. Interestingly, on *Dlk1* promoter we observed the high H3K27me3 enrichment and low H3K4me3 enrichment in shCtrl cells, and the opposite results were observed in Trim28 KD cells (Figure 6a,b). On IG-DMR, the H3K27me3 and H3K4me3 enrichments were increased in Trim28 KD cell. The results suggest that knockdown of Trim28 can increase active chromatin marker H3K4me3 and decrease repressive chromatin marker H3K27me3 on *Dlk1* promoter, leading to *Dlk1* gene expression.

### 2.6. Trim28 Interacts with E2f1 to Regulate Dlk1 Expression

Because Trim28 is not a DNA-binding protein, we would like to know which protein recruits it to *Dlk1* promoter. Previous studies indicated that Trim28 associated with E2f1 and E2f1 bound to *Dlk1* promoter [34,35]. Knockdown of E2f1 attenuated *Dlk1* expression and increased *Pparg* expression at treated with FDMI for two days (Figure 7a). Immunoprecipitation performed by overexpression of Flag-E2f1 and HA-Trim28 in 293T cells demonstrated the interaction between Trim28 and E2f1 (Figure 7b). The luciferase reporter assay showed higher level of E2f1 can significantly activate the *Dlk1* promoter and this activation is greater inhibited under overexpression of Trim28 (Figure 7c). The results show that *Dlk1* expression is activated by E2f1 and repressed in the presence of Trim28. 

## 3. Discussion

Our results provide evidence to support Trim28 KD impairs the differentiation of 3T3-L1 preadipocytes as previously described [36]. The RNA-seq analysis showed the effects of Trim28 KD are mainly in the cell cycle on day 0 and in PPAR signaling pathway on day 2. The knockdown of Trim28 leads to increasing the expression of cyclin A and cyclin B at day 0 (Appendix A). It is consistent with previous study that TRIM28 can interact with HP1 to silence the expression of cyclin A2 [16]. When confluent 3T3-L1 preadipocytes are induced to differentiation, they will run two to three cell cycles named mitotic clonal expansion to increase cell numbers, which is required for adipogenesis [37]. Compared to control cells, the cell numbers of Trim28 KD are decreased (Figure 1d). This result indicates regulating mitotic clonal expansion is a role of Trim28 to control adipogenesis. 

In GO biological processes, different Wnt signaling pathways were upregulated at day 0 and day 2, respectively (Appendix A). Wnt4 and Wnt5a, which are components of the non-canonical Wnt pathway, are upregulated in Trim28 KO cells on day 0 only, and they are known to function as positive regulators at initial stage of adipogenesis [38]. However, Wnt10b activating the Wnt/β-catenin pathway and maintains preadipocytes phenotype by inhibiting expression of C/EBPα and PPARγ [39], was upregulated at day 2 only. It suggests that Trim28 KD might mediate Wnt10b expression to attenuate C/EBPα and PPARγ induction and further stop adipogenesis. Moreover, Setdb1, which is associated with chromatin silencing and fat cell differentiation, is significantly upregulated in Trim28 KD cells on day 0 and day 2 (Appendix A). It was consistent with the previous report that Setdb1 inhibited adipogenesis by maintaining H3K9 methylation at the *Cebpa* and *Pparg* promoter [22]. Setdb1 was also activated by Wnt5a to suppress PPARγ expression and determine the fate of mesenchymal stem cells [40,41]. The results indicate that attenuating the differentiation of 3T3-L1 preadipocytes via Trim28 KD seems to be regulated by the cascades of repressors activation.

Dlk1 is one of the repressors activated in Trim28 KD 3T3-L1 cells at both day 0 and day 2. It was correlated with dysregulating imprinted gene networks under the haploinsufficiency of Trim28 [13]. The further knockdown of Dlk1 can rescue Trim28 KD-resulting inhibition of adipogenesis (Figure 4). Dlk1 is a transmembrane glycoprotein with an epidermal growth factor (EGF)-like repeats in the extracellular domain [42]. A soluble form of Dlk1 can be released by the TNFα-converting enzyme to and then interact with either Notch or fibronectin to inhibit adipogenesis [43,44,45] through the activation of MEK/ERK signaling [46]. The *Dlk1-Gtl2* cluster imprinted genes contain three paternally expressed protein-coding genes (*Dlk1, Rtl1, and Dio3*), multiple maternally expressed untranslated RNAs (including *Gtl2*), and three DMRs are methylated on the paternal allele [31,32,47,48,49]. In mouse embryos, Trim28 can bind IG-DMR in a methylation-specific manner [50]. A recent study about human embryonic cells showed that knockout of TRIM28 resulted in lower DNA methylation at paternal DMR and led to increases of *MEG3* (mouse *Gtl2*) RNAs [51]. Moreover, it was reported that knockdown of Trim28 increased the *Dlk1* expression via DNA methylation alteration on IG-DMR and Gtl2-DMR in sheep embryonic fibroblasts [6]. Biallelic expression of DLK1 was also observed in some human embryonic cell lines due to hypo- or hyper-methylation at IG-DMR, indicating cell culture conditions might influence genomic imprinting [52]. However, our results showed no significant changes in DMR methylation in Trim28 KD 3T3-L1 preadipocytes, maintaining a high methylation status (Figure 5). On the other hand, knockdown of Trim28 can increase gene activation marker H3K4me3 and decrease gene repression marker H3K27me3 on *Dlk1* promoter (Figure 6). Thus, the results suggest that Trim28 might involve in epigenetic regulation of *Dlk1* gene through histone modifications on *Dlk1* promoter in 3T3-L1 preadipocytes. 

It is well known that E2f1 is required for mitotic clonal expansion [53]. Its activity is controlled by associated retinoblastoma proteins and cell cycle regulators cyclins/CDKs [54,55]. How Trim28 modulates E2f1 protein complex to regulate the expression of target genes is unclear. Whether some of KRAB-ZFPs are involved in Trim28-mediated adipogenesis regulation will be investigated.

## 4. Materials and Methods

### 4.1. Plasmid Constructs

Flag-Trim28 expression plasmid was constructed as described [56], and the plasmid was digested by EcoRI/SalI, and the fragment was ligated to pCMV-HA-N vector (TaKaRa Bio, Kusatsu, Japan) for HA-Trim28 expression. The *Dlk1* promoter-driven luciferase reporter was constructed by subcloning the PCR product of nucleotides −1002 to +18 using primers: Forward CTGTCTGCATTTGACGGTGAAC and reverse CCGCCTTTTCGTACTGTC into pGL3-basic and sequences confirmed. The Myc-DDK-E2f1 was purchased from OriGene (Rockville, MD, USA). 

### 4.2. Cell Culture

3T3-L1 preadipocytes were purchased from American Type Culture Collection (ATCC, CL-173) and cultured in Dulbecco’s modified Eagle’s medium (DMEM; Gibco, Waltham, MA, USA) supplemented with 10% newborn calf serum (NBCS; Gibco, Waltham, MA, USA), 100 U/mL penicillin, and 0.1 mg/mL streptomycin (Corning, Corning, NY, USA) at 37 °C with 5% CO_2_ in a humidified incubator. The density of 3T3-L1 cells were maintained about 60% confluent to keep the preadipocytes phenotype. To stimulate the differentiation of preadipocytes, two-day post-confluent 3T3-L1 cells were treated with fresh medium containing 10% fetal bovine serum (FBS; HyClone, Logan, UT, USA) with hormonal cocktails including 5 μM dexamethasone (DEX; Sigma-Aldrich, St. Louis, MO, USA), 0.5 mM 3-isobutyl-1-methylxanthine (MIX, Sigma-Aldrich, St. Louis, MO, USA), and 5 μg/mL bovine insulin (Sigma-Aldrich, St. Louis, MO, USA). FDMI is short for FBS, DEX, MIX, and insulin. Human embryonic kidney 293T cells was cultured in DMEM (Gibco, Waltham, MA, USA) supplemented with 10% fetal bovine serum (FBS, Hyclone, Logan, UT, USA), 100 U/mL penicillin, and 0.1 mg/mL streptomycin (Corning, Corning, NY, USA) at 37 °C with 5% CO_2_ in a humidified incubator.

### 4.3. Oil Red O Staining

After inducing differentiation for 8 days, 3T3-L1 were washed with PBS carefully and fixed with 3.7% formaldehyde diluted in PBS at room temperature for 15 min. Cells were incubated with filtered Oil red O (Sigma-Aldrich, St. Louis, MO, USA) at room temperature for 30 min and then washed four times by double-distilled water (ddH_2_O). Stained cells were photographed by the microscope (Carl Zeiss Axiovert S100, Oberkochen, Germany) connected to the computer software. After that, Oil Red O was dissolved in 1 mL of 100% isopropanol and incubated at room temperature for 5 min. The triglyceride content was measured by spectrophotometer at OD 495 nm. The relative values were normalized with the control sample.

### 4.4. Nuclear Extracts Preparation and Western Blot

To prepare nuclear extracts, 5 × 10^6^ 3T3-L1 cells were resuspended in 400 µL of buffer A (10 mM HEPES pH7.9, 10 mM KCl, 1.5 mM MgCl_2_, 1 mM DTT, and protease inhibitors). The cell suspension was on ice for 15 min, and then 25 µL of 10% NP-40 was added followed by vortexing for 10 s. After centrifugation at 10,000× *g* for 30 s, the supernatant was collected as cytoplasmic extract. The nuclear pellets were resuspended in 100 µL of buffer C (20 mM HEPES pH7.9, 400 mM NaCl, 1 mM EDTA, 1 mM EGTA, 1 mM DTT, and protease inhibitors) and rocked on ice for 20 min. After centrifugation at top speed for 10 min, the supernatant was collected as nuclear extract. Protein concentration was quantified by Bradford method (Bio-Rad, protein assay kit). Four-fold sample buffer (200 mM Tris pH 6.8, 8% SDS, 0.4% bromophenol blue, 40% glycerol, 400 mM β-mercaptoethanol) was added to protein samples and samples were boiled at 100 °C for 5 min. The denatured proteins were subjected to sodium dodecyl sulfate polyacrylamide gel electrophoresis (SDS-PAGE) and transferred to 0.45 μm-pore-size polyvinylidene fluoride (PVDF) membranes (Millipores) by semi-dry transfer machine (Hoefer, Holliston, MA, USA). After blocking with 5% skim milk, membranes were probed with indicated primary antibodies (anti-TRIM28 from Biolegend (San Diego, CA, USA); anti-PPARγ and anti-Lamin A/C from Santa Cruz Biotechnology (Dallas, TX, USA); anti-HA from Bethyl (Montgonery, TX, USA); anti-Flag from Sigma-Aldrich (St. Louis, MO, USA)) for two hours at room temperature or at 4 °C for overnight. Membranes were washed with PBST (0.1% Tween-20 in PBS) for 10 min three times and then incubated with secondary antibodies conjugated horseradish peroxidase (KPL) for one hour at room temperature. Then membranes were washed with PBST for 10 min three times, and unbound antibodies were removed with the wash step. Western Lightening Plus Enhanced Chemiluminescence reagent (Perkin Elmer, Waltham, MA, USA) was added to membranes and the emission luminescence was exposed to X-ray film (Fujifilm, Tokyo, Japan).

### 4.5. Dual Luciferase Reporter Assay

After seeding the 293T cells in 12-well about 30–40% confluent (1 × 10^5^ cells) with 1 mL medium for overnight, the calcium phosphate precipitation transfection was conducted. Every well was transfected with 0.7 μg DNA (contain desired plasmids and pCMV-Renilla Luc as an internal control) in 100 μL. Cells were harvested after 24 h, and lysed in 25 μL of passive lysis buffer (Promega, Madison, WI, USA). The cell lysates were subjected to dual luciferase reporter assays following the manufacturer’s protocol (Promega, Madison, WI, USA). After subsequently adding substrates, the firefly luciferase activities and Renilla luciferase activities were measured by luminometer (Packard, Downer Grove, IL, USA). The firefly luciferase activities were normalized to Renilla luciferase activities. The relative luciferase activities represented that the luciferase activities of reporter carrying *Dlk1* promoter were normalized to the reporter only. Each treatment group contained triplicates, and each experiment was repeated at least three times.

### 4.6. Whole Cell Extracts Preparation and Immunoprecipitation Assay

293T cells had been seeded in 6-cm dishes for one day before transfection, and transfected with indicated plasmids using calcium phosphate precipitation method. Cells were washed one time with PBS and lysed with WCE buffer (25 mM HEPES pH 7.5, 300 mM NaCl, 1.5 mM MgCl_2_, 0.2 mM EDTA, 0.1% Triton X-100, 1 mM DTT, and protease inhibitors). The cell lysates were shaken at 4 °C for 30 min and centrifuged at 12,000 rpm, 4 °C for 5 min. The cell extracts were pre-cleared by using protein G agarose (Sigma-Aldrich, St. Louis, MO, USA) for 1 h at 4 °C and immunoprecipitated by using anti-Flag M2 agarose (Sigma-Aldrich, St. Louis, MO, USA ) or anti-HA agarose (Sigma-Aldrich, St. Louis, MO, USA) for 2 h at 4 °C. Unbound proteins were removed by washing with WCE buffer three times and diluted protein sample buffers were added. Samples were boiled at 100 °C for 5 min and subjected to western blot analysis.

### 4.7. Lentivirus Production and Infection of 3T3-L1 Cells

Lentiviral vector-mediated shRNA technology was used to knock down the *Trim28* and *Dlk1* genes. Two mouse Trim28 short hairpin RNAs (shRNA) target sequences #1:AGACATCGTGGAGAATTATTT (TRCN0000304660), #2: GGACTACAATCTGATTGTTAT (TRCN0000304607), Dlk1 shRNA CCATCGTCTTTCTCAACAAGT (TRCN0000095446), E2f1 shRNA CTCACTCCTGGAGCATGTTAA (TRCN0000374127) and scrambled control pLKO.1-shLuc (TRCN0000072243) were obtained from the National RNAi Core Facility at the Academia Sinica, Taipei, Taiwan. Viruses were produced by using calcium phosphate precipitation transfection. Fourteen micrograms of pPGK-GFP or pLKO.1-shLuc or -shTrim28 and shDlk1 constructs with 14 μg of pCMVΔR8.91 and 2 μg of pMD.G were co-transfected in early subculture of 293T cells. After eight hours, the medium was replaced with 3T3-L1 maintain medium to collect virus. Then, 3T3-L1 cells in 6-well plate were infected with the viral supernatants in the presence of 8 μg/mL of polybrene for 48 h and further selected with 3 μg/mL puromycin for one week. The whole cell extracts and total RNA were extracted for knockdown efficiency examination by western blotting and real-time PCR, respectively.

### 4.8. DNA Methylation Analysis

Genomic DNA samples were harvested and subjected to bisulfite mutagenesis with the EpiTect Fast DNA Bisulfite kit (Qiagen). After the bisulfite treatment, genomic DNA was purified and amplified by nested PCR with the primers corresponding to the imprinted DMR regions and the *Dlk1* promoter (Appendix A). All PCR conditions were 94 °C for 30 s, 52 °C for 1 min, and 72 °C for 1 min for 30 cycles in the first PCR and 35 cycles in the second PCR [57,58]. The second PCR products were subcloned into pCRII-TOPO TA vector. Fifteen to 20 clones of each construct were selected to sequence. The CpG islands when unmethylated were sensitive to bisulfite mutagenesis, but the sequences were preserved if these CpG islands were methylated. 

### 4.9. RNA Extraction and Reverse Transcription

Cells in 6-cm dish were washed with PBS and lysed with 1 mL TRIzol reagent (ThermoFisher, Waltham, MA, USA). After incubated for 5 min at room temperature, the homogenized samples were added with 200 μL of chloroform and shaken for 15 s. The samples were incubated at room temperature for 3 min and then centrifuged at 13,000 rpm for 15 min at 4 °C. The upper aqueous phase containing RNA was collected into new vials and mixed with equal volume of isopropanol. After incubation at room temperature for 10 min, the mixtures were centrifuged at 13,000 rpm for 10 min at 4 °C to precipitate RNA. The supernatant was discarded and the RNA pellets were washede twice by 100% ethanol, and once by 75% ethanol. Then, the RNA pellets were air-dried for 30 min. RNA was dissolved with 10 μL of DEPC H_2_O. After quantification by A 260/280 measurements, 2 μg RNA was annealed with 0.5 μg oligo dT at 70 °C for 10 min and then reverse transcribed into cDNA using M-MLV reverse transcriptase (Promega, Madison, MA, USA) following the manufacturer’s instructions.

### 4.10. Real-Time PCR

Quantitative real-time PCR was performed with the Applied Biosystems 7300 Real-Time PCR System (Applied Biosystems, Foster City, CA, USA). The total volume was 20 μL including Faststart Universal SYBR Green Master (Roche, Basel, Switzerland), 20-folds diluted cDNA, and 0.3 μM forward and reverse primers as shown in Appendix A. The amplification conditions were 40 cycles of 95 °C for 15 s and 60 °C for 1 min. The results were analyzed by 2^−∆∆Ct^ relative quantitation method.

### 4.11. RNA Sequencing (Quantification)

RNA samples were prepared according to the RNA extraction protocol. All subsequent technical procedures including mRNA purification and fragment, cDNA synthesis, A-tailing and adapter ligation, PCR amplification, and Illumina sequencing were performed digital gene expression (DGE) at Genomics (Taipei, Taiwan) by using Illumina NextSeq 500 sequencing system. The resulting reads were aligned to mouse reference genome GRCm38 using Bowtie 2 [59], and the numbers of sequencing reads mapped to each gene in the reference were tabulated. Referring to the significance of digital gene expression profiles, a strict algorithm was developed by Genomics company to identify differentially expressed genes (DEGs) between two samples. The DEGs were displayed in Appendix A. 

### 4.12. Chromatin Immunoprecipitation Assay

3T3-L1 cells were seeded in 10-cm dishes with 100% confluent. Until reaching differentiation condition, cells were crosslinked with 1% formaldehyde in medium at 37 °C for 10 min. The final concentration of 125 mM glycine was added to quench unreacted formaldehyde for 5 min. The medium was and cells were washed with 5 mL cold PBS three times or more to remove as much triglyceride as possible. Cells were lysed in 300 μL of lysis buffer containing protease inhibitor cocktail (Sigma-Aldrich, St. Louis, MO, USA) on ice for 15 min, and the cell suspension was vortexed briefly every 5 min. The nuclei were collected by centrifugation at 9000× *g* for 5 min at 4 °C and re-suspended with 160 μL nuclear lysis buffer containing protease inhibitorson ice for 10 min. The nuclear lysates were sonicated on ice with water at middle strength, sonicating 30 s and stopping 30 s, for 30–60 min. Two microliters of sheared chromatin was resolved by electrophoresis through the 2% agarose gel to check the length of DNA fragments between 200–600 bp, and centrifugated at 12,000× *g* for 5 min at 4 °C. The supernatant was collected and diluted 10-fold with dilution buffer (20 mM Tris-HCl pH 8.1, 167 mM NaCl, 0.01% SDS, 1% Triton X-100, 1 mM EDTA) containing protease inhibitors. The soluble chromatin complexes were separated into two parts for IP with pre-immune serum and anti-TRIM28, respectively, and took 1% of the chromatin as input preserved at 4 °C. Then, the samples were incubated with protein A/G magnetic beads overnight after the immunoprecipitating antibodies were conjunct with 20 μL magnetic protein A/G beads for two hours. The antibodies include anti-KAP1 (abcam, ab10483), anti-H3 (abcam, ab1791), anti-H3K4me3 (abcam, ab8580), and anti-H3K27me3 (abcam, ab6002). The beads were washed with 0.5 mL of each following buffers: Low salt wash buffer (0.1% SDS, 1% Triton X-100, 2 mM EDTA, 20 mM Tris-HCl pH 8.1, 150 mM NaCl) once, high salt wash buffer (0.1% SDS, 1% Triton X-100, 2 mM EDTA, 20 mM Tris-HCl, pH 8.1, 500 mM NaCl) once, LiCl wash buffer (0.25M LiCl, 1% IGEPAL CA630, 1% deoxycholic acid (sodium salt), 1 mM EDTA, 10 mM Tris pH 8.1) once, and TE buffer (10 mM Tris-HCl pH 8.0, 1 mM EDTA) once. Each wash step was incubated for 5 min on rotating platform at 4 °C. Chromatin complexes were eluted with 100 μL elution buffer (1% SDS, 0.1 M NaHCO_3_) containing 1 μL proteinase K for 2 h at 62 °C with vortex, and then incubated at 95 °C for 10 min to de-crosslink the chromatin complexes. The samples were cooled down to room temperature, and the supernatant was transferred to the new tube. DNA purification steps were followed in the manufacturer’s protocol by using the Magna ChIP^TM^ A/G kit (Millipore, Burlington, MA, USA). The DNA was analyzed by quantitative real-time PCR followed by the previous steps. The forward and reverse primers were: *Dlk1*-F: GTGGTTTTCGTGTGTGCATC and *Dlk1*-R: AACGCTCACAGACACAGTAAG; IG-DMR-F: GGAAGACAAAGAGCAAGCCTGT and IG-DMR-R: CTAGACCAACGGTGAGCCAGGAT; *Gtl2*-DMR-F: CAAGATAGTCCGTCAGAATCGGGG and *Gtl2*-DMR-R: GGGCGATTTGTAGACAGAAACTGG [60]. 

### 4.13. Statistical Analysis

The results were presented as the mean ± standard deviation (SD) of at least three independent experiments. The statistically significant values were calculated by one-tailed student’s *t*-test, shown in Figure 1, Figure 2 and Figure 3, by 1-way or 2-way ANOVA analysis, shown in Figure 4, Figure 5 and Figure 6. The significance was labeled as * *p* < 0.05, ** *p* < 0.01, *** *p* < 0.001, **** *p* < 0.0001 or ns, not significant.

## 5. Conclusions

Knockdown of Trim28 in 3T3-L1 preadipocytes activates several repressors in adipogenesis including Wnt proteins and Dlk1, leading to differentiation inhibition. Dlk1 activation in Trim28 KD cells is correlated with histone modifications of *Dlk1* promoter but not with DNA methylation in DMRs.

## Figures and Tables

**Figure 1 ijms-21-07245-f001:**
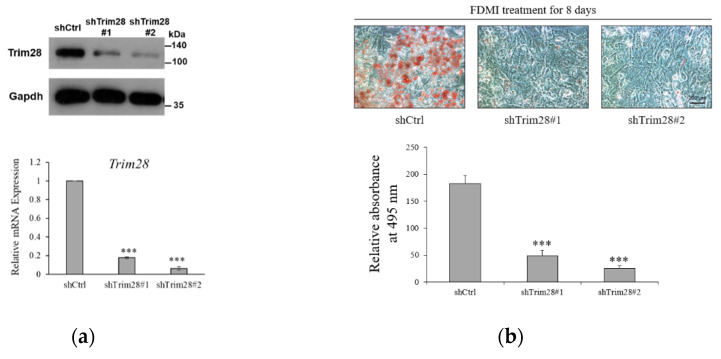
Trim28 is required for differentiation of 3T3-L1 preadipocytes. (**a**) Protein and mRNA expression of Trim28 in knockdown 3T3-L1 preadipocytes. 3T3-L1 cells were transduced with lentiviral-carrying shRNA against Trim28 (shTrim28#1 and shTrim28#2) and a non-specific control (shCtrl). (**b**) The oil red O staining of control and Trim28 KD 3T3-L1 after FDMI (FBS, DEX, MIX and Insulin) treatment for differentiation of eight days. (**c**) mRNA expression of adipogenesis markers in Trim28 KD cells during treated with FDMI for zero, one, two, and four days. (**d**) 3T3-L1 cells treated with FDMI for zero, one, two, and four days. The cells were stained with trypan blue and the cell numbers were calculated by automated cell counter. The relative cell number was normalized with control cells at 0 day. *** *p* < 0.001, ** *p* < 0.01, * *p* < 0.05.

**Figure 2 ijms-21-07245-f002:**
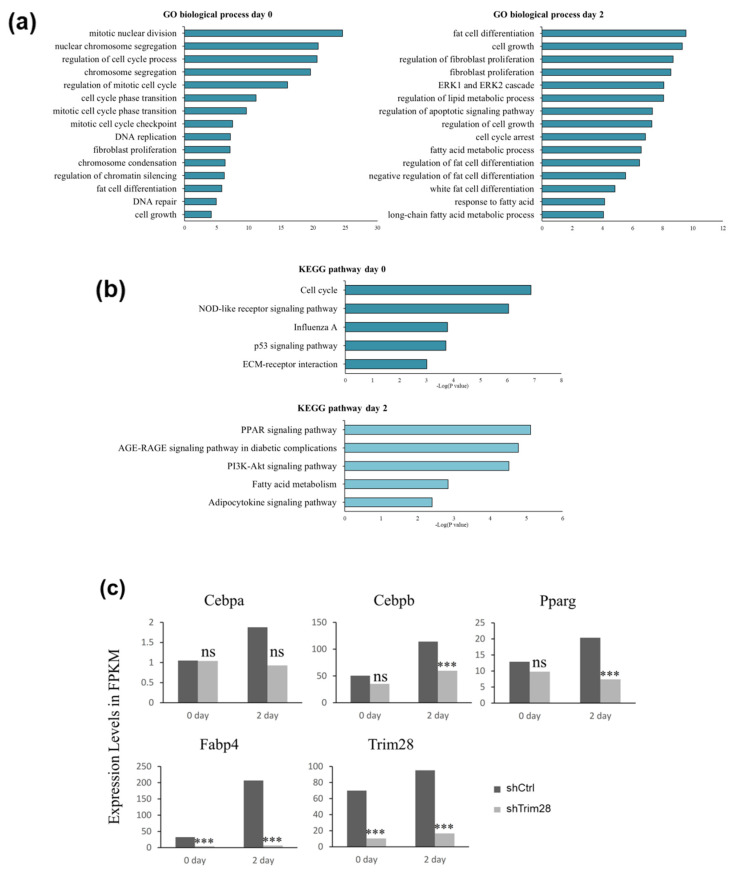
RNA-seq analysis in Trim28 KD 3T3-L1 cells. Genes were characterized as gene ontology (GO) biological processes (**a**) and Kyoto encyclopedia of genes and genomes (KEGG) pathway (**b**) in Trim28 knockdown (KD) vs. control cells at day 0 and day 2. (**c**) The expression level in FPKM (fragments per kilobase of transcript per million) of master regulators of adipocyte differentiation in Trim28 KD vs. control cells at day 0 and day 2 were shown. *** *p* < 0.001, ns: not significant.

**Figure 3 ijms-21-07245-f003:**
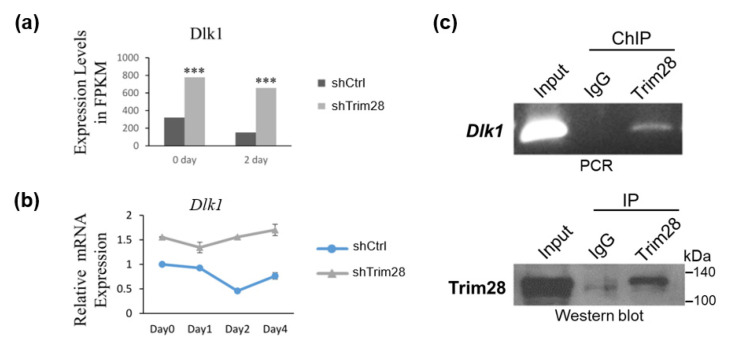
Knockdown of Trim28 increases *Dlk1* mRNA expression in 3T3-L1 cells. (**a**) *Dlk1* expression level in RNA-seq at day 0 and day 2 in control and Trim28 KD 3T3-L1cells. *** *p* < 0.001. (**b**) The kinetic of *Dlk1* mRNA expression during FDMI induction for differentiation in control and Trim28 KD 3T3-L1cells. (**c**) Chromatin-IP (ChIP)-qPCR. ChIP was performed in day 0 3T3-L1 cells by anti-Trim28 or control IgG. The precipitated genomic DNA was analyzed by semi-quantitative PCR with *Dlk1* primers positioned at −481 to −341 bp upstream of the transcriptional start site (TSS). 1% volume of each cell lysate was analyzed by PCR to determine the input signals. The IP process was also monitored by western blot as shown in lower panel.

**Figure 4 ijms-21-07245-f004:**
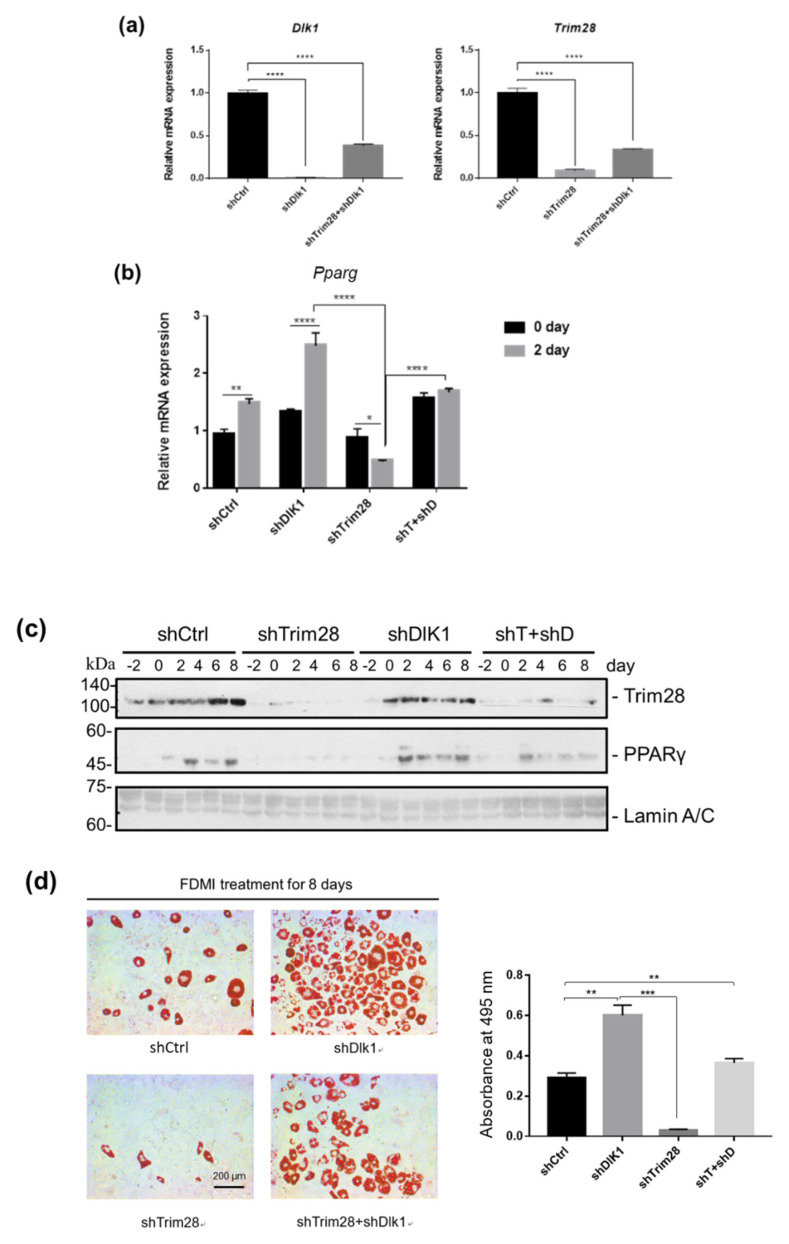
Knockdown of Dlk1 in Trim28 KD 3T3-L1 rescues cell differentiation. (**a**) Real-time PCR analysis of *Dlk1* and *Trim28* mRNA expression in 3T3-L1 cells transduced with lentiviral shRNA vectors for a standard control, Dlk1, Trim28, and Trim28+ Dlk1. **** *p* < 0.0001 (**b**) Real-time PCR analysis of *Pparg* expression in 3T3-L1 cells as described in (**a**) after differentiation for zero days and two days. Quantitative PCR data were estimated by two-way ANOVA, **** *p* < 0.0001, ** *p* < 0.01, * *p* < 0.05. (**c**) Western blot analysis of nuclear extracts from indicated 3T3-L1 cells induced differentiation for zero to eight days. The Trim28 and PPARγ protein expression was detected. The expression of the 63-kDa Lamin C and 74-kDa Lamin A isoforms was used as a loading control. (**d**) Oil red O staining of the fat droplets on the indicated knockdown 3T3-L1 cells induced differentiation for eight days. Oil red O was dissolved in isopropanol consequently, and the content of dissolution was measured by spectrophotometer at OD 495 nm. *p* value was estimated by one-way ANOVA, *** *p* < 0.001, ** *p* < 0.01.

**Figure 5 ijms-21-07245-f005:**
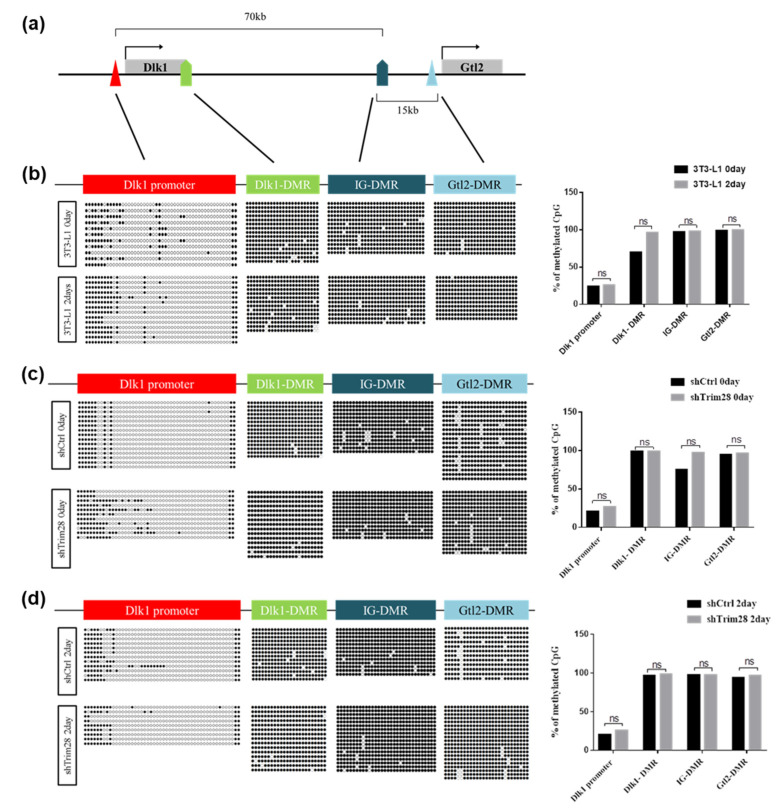
DNA methylation analysis on *Dlk1* promoter and differentially methylated regions (DMRs) of *Dlk1-Gtl2* gene cluster. (**a**) The schematic diagram of the *Dlk1-Gtl2* gene cluster. The *Dlk1* promoter, *Dlk1*-DMR, IG-DMR, and *Gtl2*-DMR were analyzed for DNA methylation. (**b**) Bisulfite sequencing of the day 0 and day 2 differentiated 3T3-L1 cells depicting the methylation status of each individual CpG island in these DMRs. The circle indicates a single CpG dyad; open circles represent unmethylated CpGs and filled represent methylated CpGs. Quantitative analysis of methylation status in total CpG dyads at *Dlk1-Gtl2* DMRs via sanger sequencing. The data were estimated by two-way ANOVA, ns indicates not significant. (**c**) Bisulfite sequencing of day 0 undifferentiated Trim28 KD cells and shCtrl 3T3-L1 cells. (**d**) Bisulfite sequencing of day 2 differentiated Trim28 KD cells and shCtrl 3T3-L1 cells.

**Figure 6 ijms-21-07245-f006:**
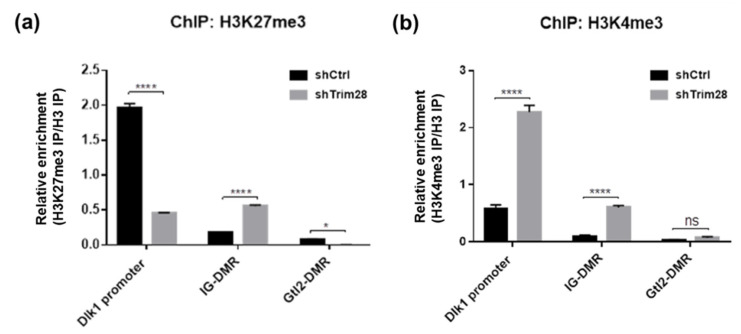
ChIP-qPCR analysis of *Dlk1* promoter, IG-DMR and Gtl2-DMR. ChIP was performed in control and Trim28 KD 3T3-L1 cells by antibodies against H3K27me3 (**a**) and H3K4me3 (**b**). The precipitated genomic DNA was analyzed by qPCR with primers as indicated. The sequences of qPCR primers for ChIP are listed in Appendix A. The relative enrichment was shown by normalized with histone H3 IP. Data were representative of three replicate experiments. The data were estimated by two-way ANOVA, **** *p* < 0.0001, * *p* < 0.05, ns: not significant.

**Figure 7 ijms-21-07245-f007:**
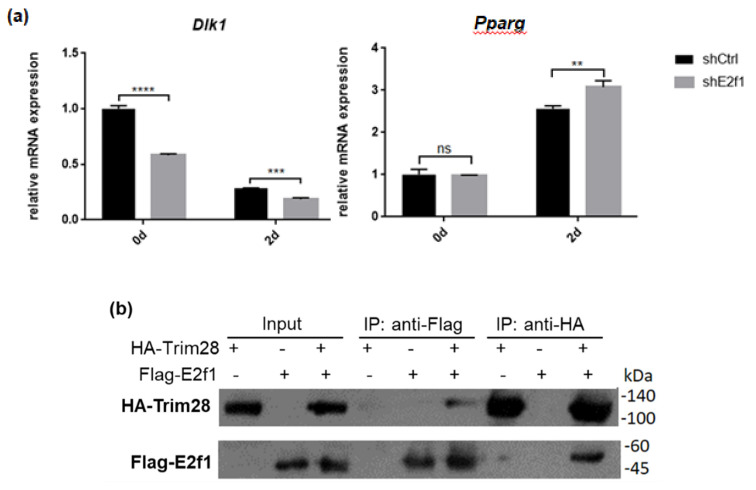
Trim28 interacts with E2f1 to regulate *Dlk1* expression. (**a**) Real-time PCR analysis of *Dlk1* and *Pparg* expression in E2f1 KD 3T3-L1 cells after treated with FDMI for zero and two days. (**b**) Immunoprecipitation of Flag-E2f1 and HA-Trim28 overexpressed in 293T cells. (**c**) *Dlk1* promoter-driven luciferase assay. The plasmids were transfected into 293T cells as indicated and cell lysates were collected for luciferase assay. The lower panel showed the protein expression level. **** *p* < 0.0001, *** *p* < 0.001, ** *p* < 0.01, ns: not significant

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
