# Peer review of "TRIM28 Regulates Dlk1 Expression in Adipogenesis"

_ijms, 2020, doi:10.3390/ijms21197245_

Round 1

Reviewer 1 Report

The manuscript by Lu et al addresses molecular pathways involved in transcriptional control of adipogenesis mediated by corepressor Trim 28. They use a well established cell model, 3T3-L1 cells, that undergo controlled adipogenic differentiation. As a summary, the authors claim (although in a very shy way) that a Trim28-Dlk1 axis regulates differentiation through Trim28 repressive recruitment to Dlk1 mediated by transcription factor E2F1.

Perhaps the reason for the weakened claim (both in abstract and in a conclusion at the end) has to do with poor experimental support.

Part of the results shown are of a rather confirmatory nature, checking out evidence already published such as stated in:
- Line 217: "Our results provide evidence to support Trim28 [knockdown] KD impairs the differentiation of 3T3-L1 preadipocytes as previously described [36]"
- Lines 56, 57 Dlk1 inhibits 3T3-L1 adipocyte differentiation, while downregulation of Dlk1 by antisense expression enhances adipogenesis

In this regard, most data in Fig.1 and 2 could have been substituted by measuring Dkl1 mRNA levels in cells with downregulated Trim28 levels.

Once that it seems that impaired differentiation under reduced levels of Trim28 occurs at the same time that inhibitor of adipogenesis, Dlk1, is derepressed (although changes in mRNA levels are seemingly modest) the authors try to link molecularly both events. Evidence for chromatin association of Trim28 to the Dlk1 promoter is shown (Fig. 3c). Unfortunately, it is not possible to determine whether the extent of the association shown is of any relevance to the regulation of Dlk1.

Next, single- and double-downregulation (lentivirally-mediated expression of shRNAs) of Trim28 and Dlk1 are done assuming that rescuing Trim28 KD-induced defective repression of Dkl1 (by concurrent Dlk1 KD) would restore differentiation. However, I think that the combination of two negative effects complicates the analysis. It is possible that the outcome is sensitive to differences in downregulation efficiencies. It would have helped that Dlk1 protein levels were determined. Given that decreased Dlk1, per se, promotes differentiation, it is not that unexpected that in the double KD, were Dlk1 downregulation to be very efficient, a dominant effect over Trim28 KD is observed.

Then, an analysis of the Dlk1 locus, looking at a few epigenetic modifications (DNA methylation, histone H3 marks) is carried out. Why are K27me3 and K4me3 modifications of histone H3 chosen to tell about transcriptional status of the Dlk1 promoter? What are relative H3 methylation intensities (Fig. 6a)? That the extent of H3K27me3 modification of the Dlk1 promoter decreases with Trim28 KD might be expected given the positive effect of EZH2 H3K27 methyltransferase on adipogenesis (line 51). Why H3K9me3 was not analysed?

Finally, reasoning that because Trim28 and DNA-binding protein E2F1 interact and that E2F1 reportedly binds Dlk1 promoter, the authors attempt to establish that Trim28 repression of Dlk1 occurs upon its recruitment to the promoter through E2F1. Their experiments do not show such a thing. On the one hand (Fig. 7a), studying the effect of manipulation of E2F1 levels on Dlk1 expression (and PParg) if at all validate (or not) previously published information. On the other (Fig. 7c), measuring the activity of a reporter gene in a transient transfection assay co-expressing Trim28 and E2F1 show results that are not consistent with the expected repressive functions: luciferase activity is lower in single Trim 28 or E2F1 transfections than when expressed together.

Minor:
Fig. 1a, X-axis legend?

Fig.2c, SD? SEM? how many replicates?

Reviewer 2 Report

The manuscript entitled “TRIM28 regulates Dlk1 expression in adipogenesis” by Lu et al. described the function of Trim28 on adipogenesis. They clearly showed that Trim28 regulates the expression status of Dlk1 with changing chromatin structure at the promoter region of the Dlk1 gene. The authors also found that E2f1 contributes to this chromatin structure change.

Broad comment:

This article clearly described how Trim28 regulates the adipogenesis on the 3T3-L1 cell line. The experiments are well designed to address their questions.

Minor comments:

  1. Histone modifications should be indicated in full. For example, “H3K4” is not clear if this is H3K4me1, H3K4me2, or H3K4me3. As you know, all modifications have different meanings.
  2. Some bar graph figures do not have error bars.
  3. In figure 2, since this is RNA-seq data, the adjusted (FDR) p-values should be used.
  4. The description of RNA-seq is not clear enough to reproduce their findings. The authors need to describe the reference assembly version, the name of the aligner they used. Moreover, the analysis method of finding differentially expressed genes should be described.  
  5. Data availability is not described (e.g., a public depository for the RNA-seq data).

Reviewer 3 Report

We investigated the role of TRIM28 in adipogenesis and its underlying mechanism. siRNA-mediated knockdown of TRIM28 lowered intracellular lipid accumulation and the expression of adipogenic genes. RNA-seq analysis demonstrated that mRNA level of Dlk1 (Pref-1) was increased in Trim28-knockdown 3T3-L1 cells. Moreover, the H3K4 methylation in the promoter region of Dlk1 was increased, but H3K27 was decreased in Trim28-knockdown cells. Furthermore, Trim28 binds with E2F1 to regulate Dlk1 expression, which was demonstrated by IP study. The results are sound. However, some additional studies are needed to mention the current conclusion. There are concerns that should be addressed. Moreover, English should be improved. There re typos in the text.

  1. What means “100%” in triglyceride content? Moreover, the authors expressed Triglyceride content by measuring isopropanol-eluates at 495 nm. This is incorrect. Isopropanol-eluates contain many lipids except for triglyceride, but of course triglyceride is major lipid in adipocytes. The expression should be improved. “Absorbance”, but not “Triglyceride content” is better.
  2. The source of 3T3-L1 cells should be shown.
  3. The method for protein preparation should be indicated.
  4. Immunoprecipitation study was done by overexpression system using 293T cells. IP study using overexpression system often shows non-specific binding. This binding of Trim28 with E2F1 can be detected in undifferentiated 3T3-L1 cells?
  5. The effect of TRIM28 and E2F1 on the DLK1 promoter activity shown by Luc assay was performed by overexpression system. It should be shown in undifferentiated 3T3-L1 cells. This concern is raised by the same reason as described in comment #4.
  6. DNA binding of E2F1 to the DLK1 promoter is changed by suppression of TRIM28 in 3T3-L1 cells?
  7. English should be improved. There are typos. Prior to submission, the text should be carefully checked.

Reviewer 4 Report

ijms-928034

Hsin-Ping et al. shown how TRIM28 is able to regulate the adipogenesis process through Dlk1 in a well-stablished in vitro cell culture model (3T3-L1)

The manuscript is well written, structured, discussed and the aim and the conclusions are clear.

However, there are important points that need to be clarified.

1.-Figure 1C It is necessary to show the mRNA expression of Trim28, Cebpa/b; Ppasg and Fabp4, with or without knockdown Trim28 along the adipogenesis process (until day 8). This point is relevant because TRIM28 protein levels are clearly induced at the end of the adipogenesis (figure 4 C) indicating a provably more important role in mature adipocytes.

2.-Figure 2 C. Introduce error bars in the columns.

  1. Figure 3, Introduce error bars in the columns in the Figure 3A. It is necessary to explain better Figure 3C, the CoIP. It is not clear if it is performed in 3T3-L1, overexpressed, in which differentiation day, against Trim28 or a tag (HA/Flag) it is not shown inner control levels for this experiment (in a similar way that if it is well done in the figure 7b).

4.-Figure 4C. Explain the reason to use nuclear extracts and show by IF the Trim28 subcellular localization.

5.-Figure 7 B and C. It is necessary to perform these experiments in 3T3-L1, not in 293T

Minor points and Misspelling

Check the abbreviation the meaning for the statistically significance in the figure legends. Sometimes it is written “**** indicates p <0.0001” but in the figure appear just one or two asterisks. Additionally, sometimes p is in capitals and others not.

Please, check the border columns colours in the figure 7. They are different than in the other figures.

It is necessary to introduce an extra space in line 150 (between “Lamin” and “A”) and in line 153 between “495” and “nm”

There is an extra point at the end of the line 215.

Line 102. Please write vs. in italics  (vs.).

Round 2

Reviewer 1 Report

  • The revised manuscript is, in essence, similar to the initially submitted version.
    Some language editing, the inclusion of controls, expanded explanation of procedures but no new results.
    Their rebuttal to specific points does not help, in general, to solve my doubts about the ability of the results to substantiate the conclusions. While the provided evidence (obviously) goes along a possible Trim28-E2F1-Dlk1 axis, as the authors call it, the experimental support is, in my opinion, weak.

Reviewer 2 Report

All my points are addressed.

Reviewer 3 Report

I have no more comment.

Reviewer 4 Report

Excellent work. Nothing to add.